# Discussion and Demonstration of RF-MEMS Attenuators Design Concepts and Modules for Advanced Beamforming in the Beyond-5G and 6G Scenario—Part 2

**DOI:** 10.3390/mi15070895

**Published:** 2024-07-09

**Authors:** Girolamo Tagliapietra, Flavio Giacomozzi, Massimiliano Michelini, Romolo Marcelli, Giovanni Maria Sardi, Jacopo Iannacci

**Affiliations:** 1Center for Sensors and Devices (SD), Fondazione Bruno Kessler (FBK), 38123 Trento, Italy; gtagliapietra@fbk.eu (G.T.); giaco@fbk.eu (F.G.); m.michelini-1@studenti.unitn.it (M.M.); 2Consiglio Nazionale delle Ricerche (CNR)—Institute for Microelectronics and Microsystems (IMM), 00133 Roma, Italy; romolo.marcelli@cnr.it (R.M.); giovannimaria.sardi@cnr.it (G.M.S.)

**Keywords:** RF-MEMS, attenuators, low actuation voltage, beamforming, 5G, B5G

## Abstract

In this paper, different concepts of reconfigurable RF-MEMS attenuators for beamforming applications are proposed and critically assessed. Capitalizing on the previous part of this work, the 1-bit attenuation modules featuring series and shunt resistors and low-voltage membranes (7–9 V) are employed to develop a 3-bit attenuator for fine-tuning attenuations (<−10 dB) in the 24.25–27.5 GHz range. More substantial attenuation levels are investigated using fabricated samples of coplanar waveguide (CPW) sections equipped with Pi-shaped resistors aiming at attenuations of −15, −30, and −45 dB. The remarkable electrical features of such configurations, showing flat attenuation curves and limited return losses, and the investigation of a switched-line attenuator design based on them led to the final proposed concept of a low-voltage 24-state attenuator. Such a simulated device combines the Pi-shaped resistors for substantial attenuations with the 3-bit design for fine-tuning operations, showing a maximum attenuation level of nearly −50 dB while maintaining steadily flat attenuation levels and limited return losses (<−11 dB) along the frequency band of interest.

## 1. Introduction

Variable attenuators are fundamental components of modern Radio Frequency (RF) systems to modify the signal power while maintaining signal integrity. As compared to variable amplifiers, variable attenuators provide higher linearity and low power consumption [1]. Among their common applications, it is worth mentioning broadband vector modulators [2] and impedance matching networks [3], measurement equipment [4], and automatic gain control amplifiers of RF front ends and transceivers [5]. In such contexts, variable attenuators may adjust or modulate the signal strength (RF applications) [6] or isolate ports and parts of the circuitry that are sensitive to a specific amount of power (duplex systems and measurement equipment) [7].

In the current telecommunication scenario, variable attenuators play a crucial role in the Multiple-Input-Multiple-Output (MIMO) systems [8], on which radio access to the 5G network is based [9]. In fact, in both base stations and small cells [10], the use of such phased-array antenna systems is meant to improve communications using beamforming, which consists of the modification of the radiation pattern of the whole antenna system. Beamforming is operated by varying the amplitude and the phase of the signal fed to each radiating element of the phased array [11]. While phase shifters are crucial to controlling the orientation of the radiation pattern, attenuators are crucial to controlling its shape [12].

A proper beamforming is crucial on the receiver’s side to reduce the impact of interfering signals coming from other directions [13], which translates into, a radiation pattern characterized by a main and narrow radiation lobe, with sidelobes of minimal amplitude. Arrays with a uniform element amplitude display a high sidelobe level (SLL) (i.e., −13.2 dB [14]) so that the tapering of the element amplitude is introduced to reduce the SLL [15]. Wide tuning ranges are required for a substantial SLL reduction, whereas fine-tuning resolutions are required for low quantization errors due to the tapering [16]. As an example, a tapering range of 21 dB is necessary to enforce a −30 dB SLL reduction according to Taylor’s tapering method [17]. In addition, a high number of radiating elements implies the need for components characterized by reduced power consumption [18], minimal losses, and high miniaturization due to the small wavelengths [19,20].

Variable attenuators realized using RF Micro-Electro-Mechanical Systems (RF-MEMS) are gaining increasing popularity since the abovementioned features can be effectively addressed by such technology [21,22,23]. Digital [24] and analog [16] architectures have been explored for the developed RF-MEMS attenuators, with digital being the predominant choice [25]. Key sought-after attributes for these devices include reconfigurability, broadband performance, and excellent linearity [26]. Efforts in research have focused on expanding the range of attenuation levels while ensuring consistent impedance characteristics across various frequencies and minimal deviation from the desired attenuation level.

Concerning the related scientific literature, most of the RF-MEMS attenuators rely on Coplanar Waveguide (CPW) transmission lines featuring metallic membranes actuated by the electrostatic principle. The membranes generally connect or disconnect the signal line to resistive loads, such as series, shunt [27], or other configurations of resistors (e.g., T- or Pi-shaped [28,29]), in devices consisting of single or multiple cascaded units [30]. Sometimes, units consisting of different signal lines, generally an unloaded line and a line equipped with fixed loads selected by switches, are cascaded [31]. The maximum attenuation levels achieved by the existing devices reach up to −20 [24], −45 [32], or −70 dB [29], usually by 3- or 4-bit arrangements [23,29], with some elaborate examples featuring 8 bits [32]. The achieved attenuation errors are generally small, ranging within a range of (0.2–5)% as compared to the target value [2,28]. The attenuation steps realized by the existing devices vary based on the architecture, with the analog implementations showing the smallest steps (0.2 dB in [20]) and the digital ones showing more substantial steps, such as 5 and 10 dB in [28,29], respectively. In terms of operational bandwidth, many examples cover the (0–20) GHz interval [28], while others touch 40 GHz [33] or even 80 GHz [32], demonstrating a propensity for millimeter-wave (mm-wave) applications.

Despite such valuable features, existing and general-purpose devices show two restrictions: in some cases, a large footprint, and always a substantial bias voltage. In fact, the pursuit of devices with many attenuation states led to footprints reaching 2.15 × 7.5 mm^2^ [24] or 8.28 × 2.37 mm^2^ [30] that can hardly fit in practical mm-wave MIMO systems (e.g., the footprint of the radiating elements of [34,35]), while the integration of devices with driving voltages of tens of Volts with common components implies the use of step-up converters, increasing costs, the design complexity, and the footprint of such systems. In this regard, commercial RF-MEMS variable attenuators often have reduced bias voltages (e.g., 3.3 V in [36]).

In this paper, we propose a compact 3-bit attenuator module for beamforming applications in the (24.25–27.5) GHz range, for fine-tuning attenuations (up to −10 dB) and low driving voltages, and a concept of a switched-line attenuator module for more substantial levels, which can be complementary to the former to expand the maximum achievable attenuation. Capitalizing on the previous part of this work [37], describing basic building blocks of cells based on series and shunt resistors and membranes with low actuation voltage, a compact layout featuring a driving voltage of 7 V and switched resistors in a sort of Pi-shaped deployment is developed and critically assessed. Moreover, fabricated samples of fixed Pi-shaped resistors targeting higher attenuation levels (−15 dB, −30 dB, and −45 dB) have been characterized, together with a design of a 2-bit switched-line attenuation module employing such resistors and operated by a larger driving voltage (~45 V), which suggested its potential to expand the capabilities of the previous module upon proper design refinement. In fact, the coarse design of the geometrical discontinuities of such a switched-line attenuator module (i.e., the T-junctions and the 90° bends) limited its employment to frequency bands up to 20 GHz. However, the potential of such a switched-line design is corroborated by the subsequent improved design, characterized by an optimized Single-Pole-Double-Throw (SPDT) with a 7 V actuation voltage that is combined with the initially described 3-bit attenuator in a simulated arrangement featuring 24 possible attenuation levels, reaching −48.14 dB without any significant degradation of the overall electrical features of the resulting device. The proposed device combines different advantages, ranging from the reduced actuation voltage up to the remarkable reconfigurability, simultaneously providing fine-tuning capabilities and substantial maximum attenuation levels.

This paper is organized as follows: A brief description of the electromechanical features of the membranes considered for the development of the 3-bit compact attenuator is reported in Section 2, while in Section 3, the development and simulated performance of the proposed 3-bit attenuator are assessed in relation to the real case scenario. In Section 4, samples of Pi-shaped resistors are considered for substantial attenuation levels, together with the concept of a 2-bit switched line module adopting such Pi-shaped resistors. Finally, Section 5 describes the combination of the proposed modules in a single conceptual integration of low-voltage multi-state attenuators, while final remarks and considerations are provided in Section 6.

## 2. The Low Pull-in Membranes

The selected class of membranes utilizes meandered supports to lower the spring constant in the vertical direction for the movable structure, thereby decreasing the required actuation voltage (or pull-in). Among the most fundamental layers that can be traced in the layouts reported in Figure 1, it is possible to notice the Polysilicon (in red) forming the decoupling resistors and the buried electrodes under the square areas of the movable membranes. It is also possible to notice the interrupted sections of the RF signal line, realized by the buried multi-metal layer (in blue), which also constitute the square pillars (Figure 1d) that are meant to stop the membrane upon the actuation and prevent direct contact between the membrane and the underlying exposed electrodes. Rectangular areas of evaporated gold are deployed in correspondence with the interrupted sections of the RF signal line (Figure 1d), forming the landing areas onto which the membrane will establish ohmic contact after the actuation. Concerning the movable membranes, a first layer of electroplated gold (Au) forms the movable structure, whereas a second and thicker layer of gold provides an enhanced stiffness to the anchor points and the central plate, as visible in Figure 1b.

The chosen type of membrane features beams with uniform meanders, except the terminal beam, which has been extended in two variants to achieve a gradually reduced pull-in voltage. In fact, the three considered variants have in common a suspended structure with the same footprint (780 × 270 μm^2^), but while the first variant (Dev1) is characterized by uniform meanders (Figure 1a,b), the terminal meander of the second (Dev2) and third (Dev3) variants has been stretched to achieve a reduced pull-in voltage without increasing the overall area. The dimensional features of the membranes are highlighted in Figure 1 and summarized in Table 1.

The discussed membranes have been employed in the simple Single-Pole-Single-Throw switch configuration of Figure 1 as their simplest configuration, and the measurements on their fabricated samples have been compared to the simulation outcomes displayed in Figure 2, obtained by the Finite Elements Method (FEM) in the Ansys Workbench software environment (2022R2 version). The measurements were conducted by measuring the resistance drop along the RF signal line, passing from a substantial value (open circuit) to a minimal one (closed circuit) upon the actuation of the membrane.

Concerning the comparison displayed in Figure 2, it is possible to notice a good agreement between the simulated and measured electro-mechanical behavior of the devices. In fact, considering that the driving voltage has been provided by 1 V steps, the measured behavior of Dev2 coincides with the simulated one, while the measured pull-in of Dev1 and Dev3 (7 V and 9 V, respectively) is moderately higher than the simulated ones (5 V and 8 V, respectively). More detailed considerations concerning the abovementioned membranes can be found in previous and more focused works [38].

## 3. Low Pull-in Series and Shunt Attenuation Cells

In the previous part of this work, we explored the potential of basic attenuation modules equipped with the discussed class of membranes characterized by a low pull-in voltage. In particular, single modules featuring series and shunt resistors as the ones in Figure 3a,b have been critically addressed, showing that the series modules could be effectively adopted to achieve attenuation levels of about −3 dB, whereas the shunt modules can introduce attenuation levels up to nearly −5 dB. However, due to their overall compactness, they could be the basic building blocks of more complex multi-state attenuators.

The working principle of those kinds of attenuation cells is demonstrated by a circuital simulation in Cadence AWR Design Environment 22.1, where a section of CPW line is loaded by a resistor, series, or shunt, Figure 4a,b respectively. The CPW line has a nominal length of 5 mm, and the resistor is considered ideal: zero length and without any imaginary part (i.e., neither capacitive nor inductive behavior). The activation of the resistor is commanded by the movement of the membranes. The effects in terms of parasitic impedances due to the geometry of the MEMS structures are not gathered, by purpose, in these simulations, which are devoted to model and analyzing the behavior of the series or shunt resistors configured with the CPW lines. The choice of using two shunt resistors is motivated by having a symmetric configuration along the longitudinal axes of the device that facilitates the control of the input impedance of the device. Different geometrical arrangements and more sophisticated configurations can be considered based on the specific requirements tailored to the overall system performance. Since the beginning of the overall design, the final device is composed of multiple cells, where the effects of parasitic impedances and eventual coupling effects can be modeled only by full-wave simulations of the entire device. The conceptual design and the global optimization of the complete design are described in Section 5.

As a proof of concept for the working principle, we show the behavior in terms of Insertion Loss (S_21_ parameter) as a function of the value of the resistor, for the two configurations described before. The results for the series configuration are reported in Figure 5, while the results related to the shunt configuration are displayed in Figure 6. It is observed that the different values of the resistor allow the tuning of the attenuation level of the CPW section. According to those operating principles, the resistors can be designed for on-wafer/on-chip realization.

In order to show the potential of the abovementioned 1-bit attenuation cells characterized by a low actuation voltage introducing a fine-tuning in the range of few dBs, a reconfigurable 3-bit attenuator has been designed and optimized. As visible in Figure 7a, it is composed by three cascaded cells, whose spacing between the cells (70 µm) and between the cells and the bordering ground planes (25 µm) has been optimized for the frequency range of interest. The same applies to the displayed CPW line, whose width and gaps are 79 µm and 46 µm, respectively. Concerning the employed membranes, the model characterized by a 7 V pull-in (Dev2 in Figure 2) has been chosen since it represents a good trade-off between its electromechanical and electrical performances. The resulting device exhibits remarkable compactness, having an overall footprint of 1.53 × 1.5 mm^2^.

Regarding the operational bandwidth of the device, the 24.25–27.5 GHz frequency interval has been selected as a practical case, since it is the portion of the spectrum allocated to 5G communications in Europe (Frequency Range 2, N258 band).

Concerning the theoretical foundation of the device developed by combining such attenuation cells, it is worth noticing that no specific formulation devised so far applies to the present case. In fact, if the design rules for fixed Pi-shaped resistors on the basis of the desired attenuation level have been outlined in [29,30], the case of individually switched resistors forming an overall Pi-shaped configuration introduces additional reactive phenomena that are not taken into account in the abovementioned formulation. Therefore, the present layout can be considered a generic lossy transmission line (widely treated in [39]), loaded by (both series and shunt) switched resistive loads and variable capacitors accounting for the capacitive couplings introduced by the membranes in rest position.

The initial guideline chosen during the optimization of the device was to achieve for each resistor a substantial value of attenuation, of about −3 dB, possibly maintaining an overall symmetry between the two lateral shunt cells. However, the optimization process highlighted the need to reduce the amount of attenuation introduced by the shunt resistors in correspondence with the input port, to mitigate the amount of the reflected power when the closest shunt cell is activated individually or in combination with the series one. As a result, the shunt resistors close to the intended input port have been dimensioned differently (lower pair in Figure 7b, whose equivalent resistor is on the left in Figure 7c), making such device electrically asymmetric. In fact, while the shunt resistors close to the input port have an 80 × 20 µm^2^ (length × width) area, the other shunt ones measure 71 × 36 µm^2^, and their sheet resistances are 60 and 190 Ohm/Sq, respectively. For the sake of completeness, 80 Ohm/Sq sheet resistance has been chosen for the series resistor, resulting in a 103 × 117 µm^2^ solid.

The following considerations about the electromagnetic behavior of the device are based on the simulation outcomes obtained by the Finite Elements Method (FEM) in the Ansys High-Frequency Structure Simulator (HFSS) software environment (2022R2 version). Concerning the electrical features reported in Figure 8a, it is possible to notice that the attenuation levels realized by the device cover and interval are about 9 dB, from the basic −1.06 dB insertion loss (when no cell is active) up to −10.15 dB, which is achieved by activating all three cells. Other interesting states are the ones in which the series resistor in combination with the left or right shunt is active (−7.10 and −6.67 dB, respectively), and the ones in which the single series or the right shunt couple of resistors is active (−3.34 and −3.05 dB, respectively).

The corresponding return loss curves in Figure 8b are generally satisfactory, except for the higher one that is related to the S21 curve realizing a −3.43 dB attenuation at 25.87 GHz. This is the case in which the only left shunt cell is active. In this regard, the impact of such a shunt cell on the return loss curves of the devices can be minimized by further reducing its attenuation. Otherwise, such a problematic state can be avoided since the device in its current layout already features states attaining attenuations close to that value, as visible in the following Table 2.

Among the different considered states, it is worth mentioning the state in which attenuation is introduced by the sole insertion loss of the device (000 state in Table 2). In this state, the central resistor is short-circuited by the actuated membrane, while the shunt membranes are in the rest position. In this situation, the only losses experienced by the signal are due to the small series contact resistance introduced by the actuated central membrane, summed to the capacitive coupling (towards the ground planes) introduced by the two resting shunt membranes. Due to the frequency-independent behavior of the contact resistance (which mainly affects the insertion loss of devices), in the 000 state, the device can be modeled as a single equivalent shunt switch in the rest position, whose return loss depends only on the up-state capacitance. In fact, the slope of the return loss curve of the 000 state in Figure 8b agrees with the one of a shunt switch in the rest position [40]. The present layout of the 3-bit reconfigurable attenuator suggests that series and shunt cells can be effectively combined to achieve a fine-tuning attenuation in the range between −1 and −10 dB. If such a tuning range must be enlarged up to more substantial base values (e.g., −20, −30, or −40 dB), the present layout can be combined with other combinations of resistors, since the compactness of this layout would not affect sensitively the footprint of the resulting device. At the same time, this 3-bit attenuator could find application in the beamforming of current antenna arrays. As an example, the tapering applied to the radiating elements of the array reported in [41] required normalized excitation amplitudes of 1, 0.9, 0.72, 0.51, and 0.4 for its radiating elements, which correspond to normalized power amplitudes of 0, −0.91, −2.85, −5.84, and −7.95 dB. Such values are quite close to the attenuation levels achieved by the proposed 3-bit attenuator, and even smaller attenuations are needed for the 4 × 4 and 8 × 8 arrays reported in [42]. This confirms that, after proper optimization, this reconfigurable attenuator could be employed in real case scenarios.

## 4. Pi-Shaped Attenuation Cells and Concept

As previously mentioned, apart from cascading multiple series and/or shunt cells, in cases where substantial attenuation values are required, an advisable solution consists of the choice of other combined topologies, like T-shaped or Pi-shaped resistors [18]. For this purpose, a basic cell featuring Pi-shaped resistors inserted along a plain CPW has been designed on the basis of the following design guidelines outlined in [29], dimensioning the series resistor (*R_series_*) and each of the four shunt resistors (*R_shunt_*) for an attenuation level of −15 dB.
(1)$$R_{shunt}=\frac{\left(10^{\frac{A}{10}}-1\right)\cdot Z_{0}}{10^{\frac{A}{10}}+1-2\cdot 10^{\frac{A}{20}}}$$
(2)$$R_{series}=\frac{Z_{0}}{2}\cdot \frac{10^{\frac{A}{10}}-1}{10^{\frac{A}{20}}}\;$$

After an initial simulation, an optimization aimed at achieving a more precise attenuation level of −15 dB was performed, and the devices featuring the optimized resistors were fabricated and characterized. As visible in Figure 9a, a 2 mm-long CPW with multi-metal underpass has been considered as basic model on which a single or multiple compounds of 15 dB Pi-shaped resistors have been placed. Concerning the optimized 15 dB Pi-shaped resistors, they are characterized by a 150 Ohm/Sq sheet resistance, with the series one being 25 × 106.5 µm^2^, while the shunt ones being 67.5 × 25 µm^2^.

Samples of plain CPW and CPW equipped with one, two, and three equally spaced Pi-shaped resistors have been fabricated in a unique batch, and their measurement outcomes are depicted in Figure 9b,c. Regarding the comparison between the plain CPW and the CPW equipped with a single Pi resistor in Figure 9b, it is worth noticing the stable flatness of the reached attenuation, ranging between −12.7 and −13 dB along the 40 GHz interval. In addition, a limited return loss (<−12 dB) can be seen along the whole frequency range. The small discrepancy between the desired (−15 dB) and the achieved attenuation level (~−13 dB) has repercussions on the attenuation levels obtained by cascading the Pi resistors. In fact, as visible in Figure 9c, two cascaded Pi resistors determined an attenuation of about −25 dB, while the three resistors in Figure 9a led to an attenuation of about −37 dB. Besides the noise ripples affecting the measurements related to the latter case, these two cases also demonstrated remarkably flat attenuation levels and limited return losses of about −15 dB.

The broadband and stable behavior of such Pi-shaped resistors at high attenuation levels make them a suitable candidate for reconfigurable RF-MEMS attenuators, even when combined with the previous compact attenuator based on series or shunt resistors. In fact, in more complex attenuation networks, the Pi-shaped resistors could be used to set a substantial and coarse attenuation, while additional series or shunt cells could be used for fine-tuning.

The abovementioned Pi-shaped resistors have been adopted to develop the attenuation cell reported in Figure 10a as an initial attempt to employ such a configuration of resistors in a reconfigurable device. This attenuation cell consists of an upper branch loaded with the Pi resistors and a lower, unloaded branch, both selectable by the respective couple of membranes. Concerning the employed membranes, as for some cells detailed in the previous part of this work, a legacy design has been considered, characterized by a more substantial pull-in voltage (~ 45 V) [27]. By such a layout of the attenuator cell, three useful states can be achieved: the one with the reference unloaded path active, the one with the loaded path active, and the one in which both are active, resulting in the equivalent resistance among the two. This is clearly visible in Figure 10b, where the appreciable attenuation level introduced by the loaded branch is depicted (−13.89 dB at 15 GHz), in combination with the insertion loss of the unloaded branch (−3.82 dB) and the attenuation achieved by activating both (−8.3 dB).

It is worth noticing that the stable behavior of the unloaded branch mitigates the ripples of the loaded branch when it is activated in parallel. On the other hand, such ripples (deviating from the flatness observable in Figure 9b) and the non-negligible insertion loss of the unloaded branch can be attributed to a coarse design of the CPW structure. In this regard, the 90° bends of the two branches and T-junctions at both ends should be subject to proper refinement, in order to minimize the undesired effects deriving from such discontinuities, which are exacerbated at higher frequencies. In fact, as known from the scientific literature [43,44], geometrical discontinuities such as T-junctions and 90° bends should be carefully designed to minimize the impact of the undesired reactive and frequency-dependent phenomena (increased inductance, capacitive couplings, and thus losses) on the global behavior of the devices. In the case of the device reported in Figure 10a, such discontinuities have not been optimized, determining a fluctuating behavior of the attenuation curves in Figure 10b above 20 GHz. For this reason, the performance of the device has been assessed along the frequency interval, where the effect of the geometrical discontinuities does not dramatically impact the overall behavior of the device.

Despite such limitations in terms of losses, it is possible to notice the limited return losses of Figure 10c, among which the most pronounced is the one related to the parallel activation of the two branches, which suffers the consequences of the mentioned coarse design. Nonetheless, the displayed curves are steadily below −10 dB along the considered interval.

## 5. Conceptual Design of a 24-State Variable Attenuator

The switched-line attenuation module described in the previous section suggests that, under the premise of a careful design aimed at mitigating the geometrical discontinuities, the discussed Pi-shaped resistors can be effectively combined with the reported 3-bit attenuator. In such a monolithic arrangement, the switched-line module would provide the coarse and substantial attenuation, while the 3-bit module would introduce the fine tuning. For this reason, in this section, we describe a configuration of a 24-state variable attenuator based on the combination of the building blocks described in previous sections. As shown by the topology of the concept device described in the following Figure 11, it is possible to combine the switched-line attenuator based on the CPW section with a Pi resistor and the 3-bit variable one through a couple of RF-MEMS-based SPDT switches and sections of CPW lines [45].

The circuit can be subdivided into two macroblocks: the first one contains the two SPDTs through which we can select a negligible attenuation, a fixed and more substantial level of attenuation, and an intermediate attenuation, respectively, routing the signal into the unloaded CPW section, the Pi-loaded CPW section, or activating both the routes. For this purpose, two bits would govern such a stage of the device. The second block consists of the 3-bit variable attenuation cell. Therefore, the resulting device is operated by 5 control bits.

Each sub-component has its own complex impedance, as shown in Figure 12. That means that when combined, all together, the overall input impedance of the final two-port devices must be designed carefully to obtain the desired value and matched with the required ones. In this design, we consider the standard value of 50 Ohm.

As visible in Figure 12, the input impedance varies, both in the real and imaginary parts, depending on the selected attenuation level. The markers identify the center-band frequency. The effect is more evident when dealing with the 3-bit cell, while the Pi-shaped CPW section is more stable. That effect is unavoidable, depending on the actuation of the series and/or shunt load resistors in the selected state.

To optimize the final layout, as shown in Figure 13, we can define a general and simple strategy by introducing five sections of straight CPW lines with the same geometrical parameters and characteristic impedance (50 Ohm). For the purpose of the simulations, the CPW are modeled with the conductivity of the Au ($\sigma =4.1\cdot 10^{7}S/m$); this choice is reasonable since in the fabricated devices all the lines are electroplated with a final thickness of 5 µm, thus it is possible to neglect the skin effect at the operating frequencies, whose value is calculated as 0.5 µm. The lengths of those sections are the optimization variables, and the goal of the optimization is the S_11_ of the entire circuit, below −10 dB, within the bandwidth of interest (24.25–27.5 GHz). In that schematic, the functional blocks, i.e., the SPDTs, the Pi-shaped CPW, and the 3-bit attenuator, are described by the S-parameters extracted from a set of EM full-wave 3D simulations; the CPW sections (A,B,C,D,E) and the 90° bends are described by the internal model implemented in the software. The schematic is replicated for each of the states of interest, in Cadence AWR Design Environment 22.1. The lengths of the five CPW sections are defined as global variables of the optimization procedure.

For the sake of brevity, we show the results related to the following configurations:SPDTs to select the unloaded CPW section and the 3-bit variable attenuator at −1.06 dB (minimum attenuation), here referred to as Case 1;SPDTs to select the unloaded CPW section and the 3-bit variable attenuator at −10.15 dB (maximum attenuation), here referred to as Case 2;SPDTs to select the Pi-shaped CPW section with a fixed level of −45 dB of attenuation and the 3-bit variable attenuator at −10.15 dB (maximum attenuation), here referred to as Case 3;

The optimization procedure has been conducted by the built-in Pointer-Hybrid method. It is not possible to know a priori if only one global minimum exists or if the optimization problem possesses various local minima, consequently, the obtained solution may be further improved and not unique, also depending on the constraints of the optimization variables. The calculated optimal solution is based on the following values of the five CPW sections, imposed as variables in the design:CPWs “A” with a length of 1964 µm;CPWs “B” with a length of 1 µm;CPWs “C” with a length of 1 µm;CPWs “D” with a length of 844 µm;CPW “E” with a length of 3819 µm.

In the following Figure 14a, the return loss (at port 1) of the complete circuit is reported, as the output of the optimization procedure, and it shows a well-matched device since the S11 curves lie below −10 dB for the selected attenuation levels. The obtained attenuation levels are displayed in Figure 14b, defined as insertion loss, or S21, of the complete circuit.

By looking at the electrical features of this simulated arrangement, it is possible to notice that the individual and remarkable return losses characterizing the single modules (loaded and unloaded CPW, 3-bit attenuator, and SPDTs) determine an overall return loss that is steadily below −10 dB along the entire considered interval. Likewise, the flatness of the individual attenuation levels is maintained when the single modules are combined. On the other hand, it is also possible to notice that the described arrangement led to slightly increased losses. In fact, the insertion loss resulting from Case 1 in the vicinity of 25.87 GHz (−3.27 dB) is greater than the plain sum of the unloaded CPW (−0.32 dB) and the 3-bit attenuator in its state of minimum attenuation (−1.06 dB) at the same frequency. The same applies to Cases 2 and 3, where the displayed attenuation levels (−12.23 and −48.17 dB) surpass the sum of the single contributions (−10.47 and −47.03 dB), respectively.

In addition to the remarkable behavior of the displayed curves, it is worth noticing the high reconfigurability of such a device, which is able to attain attenuation levels ranging up to nearly −50 dB in 24 possible states. Despite the reasonably limited losses introduced by the combination of the abovementioned attenuation modules (that can be mitigated by a further ad hoc refinement), such simulated monolithic integration proved the potential of the described modules in providing low-voltage and multi-state RF-MEMS attenuators, providing broadband and flat attenuation levels serving the beamforming architectures of the current and future telecommunications scenarios.

## 6. Conclusions

In this paper, various ideas for reconfigurable RF-MEMS attenuators aimed at beamforming applications are presented. On the basis of the previous part of this work, the 1-bit attenuation modules, which employ series and shunt resistors along with low-voltage membranes (7–9 V), are utilized to create a 3-bit attenuator capable of finely adjusting attenuations (<−10 dB) within the 24.25–27.5 GHz range. Fabricated and tested coplanar waveguide (CPW) sections with Pi-shaped resistors, aiming for attenuations of −15, −30, and −45 dB, displayed remarkable electrical characteristics, including flat attenuation curves and minimal return losses. The adoption of such resistors in a fabricated concept of switched-line attenuator highlighted that Pi-shaped resistors in such switched configurations can effectively provide different substantial and broadband attenuation levels in the case of a proper RF design. Based on the promising electromagnetic behavior of the single modules, their integration into a single multi-state device has been simulated. The resulting 24-state device, entirely based on low-voltage membranes, comprises a switched stage devoted to the basic attenuation levels, which can be tuned by the second stage, consisting of the 3-bit attenuator considered initially. The simulated integration confirmed that high attenuation levels (nearly up to −50 dB) can be achieved by such a reconfigurable device without any significant degradation of the whole electromagnetic behavior, given a proper RF design. Since greater attention has been paid to the impedance matching among the blocks composing the 24-state attenuator (S11 < −11 dB), at the expense of the current insertion loss (−3.27 dB), further developments concerning such layout will be focused on the minimization of the overall losses of such a low-voltage reconfigurable RF-MEMS attenuator concept.

## Figures and Tables

**Figure 1 micromachines-15-00895-f001:**
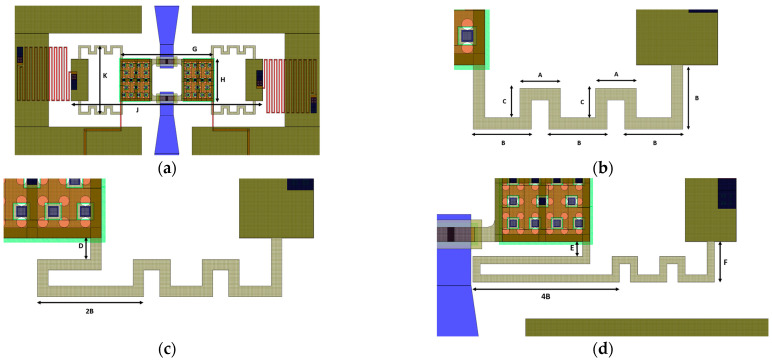
(**a**) Top view and (**b**) magnified view representing the beams of the first variant (Dev1), (**c**) magnified view of the second (Dev2), and (**d**) third variant (Dev3).

**Figure 2 micromachines-15-00895-f002:**
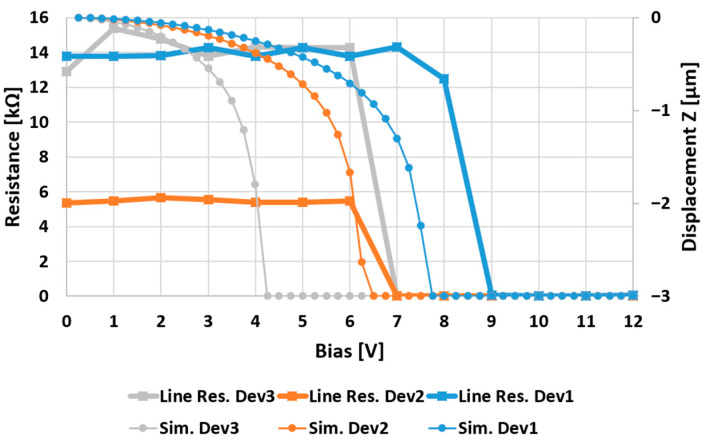
Simulated pull-in voltage of the three discussed membranes (dot marks) and their measured pull-in voltage by resistance drop along the RF signal line (square marks).

**Figure 3 micromachines-15-00895-f003:**
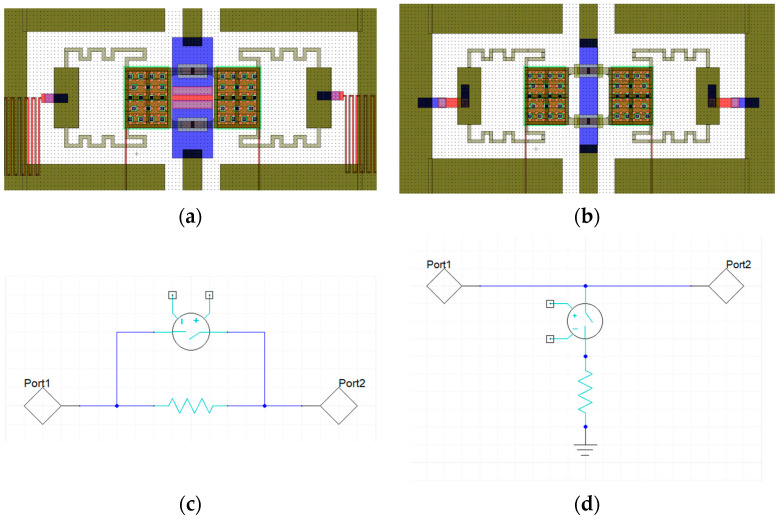
(**a**) Basic attenuator module based on a series resistor, (**b**) basic attenuator module based on two shunt resistors, (**c**) equivalent circuit topology of the series module, in which the resistor is short-circuited upon the actuation of the membrane, and (**d**) equivalent circuit topology of the shunt module.

**Figure 4 micromachines-15-00895-f004:**
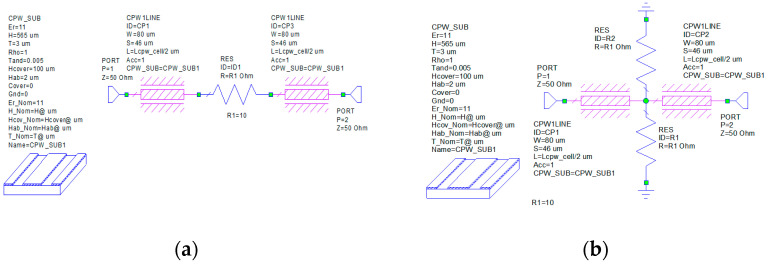
Circuital model describing the working principle of the basic attenuator module based on: (**a**) one series resistor, and (**b**) two shunt equal resistors.

**Figure 5 micromachines-15-00895-f005:**
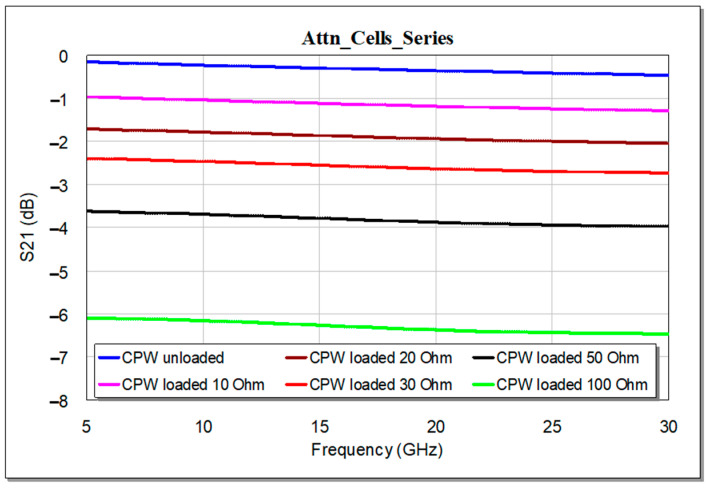
Behavior of the Insertion Loss for the CPW loaded by a series resistor, as a function of the resistor value.

**Figure 6 micromachines-15-00895-f006:**
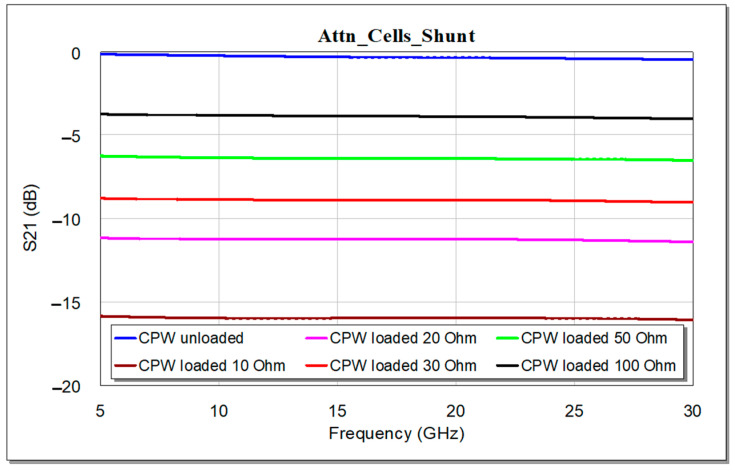
Behavior of the Insertion Loss for the CPW loaded by two equal shunt resistors, as a function of the resistor value.

**Figure 7 micromachines-15-00895-f007:**
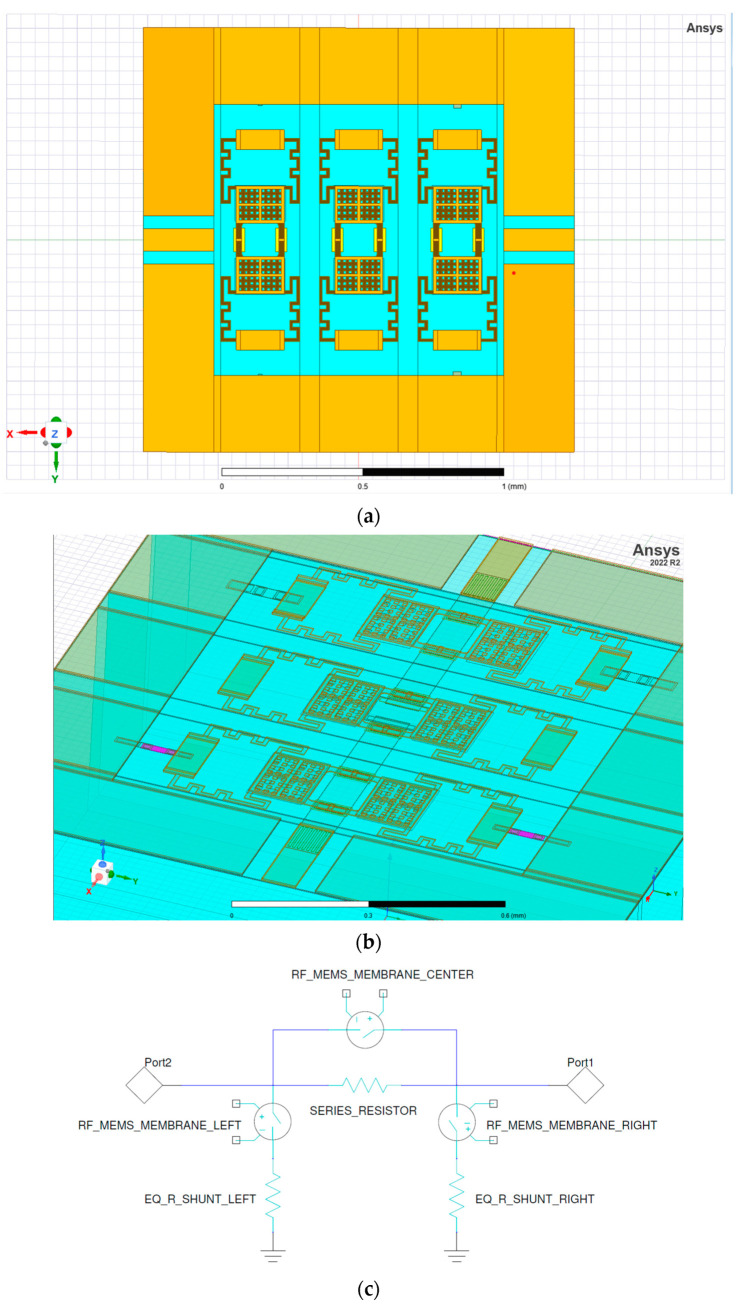
(**a**) Top view of the 3-bit reconfigurable attenuator, featuring a central series cell and lateral shunt cells; (**b**) detail of the optimized resistors, with the thinner ones (left cell) neighboring the input port; and (**c**) its equivalent circuit topology (not accounting, by purpose, parasitic impedances).

**Figure 8 micromachines-15-00895-f008:**
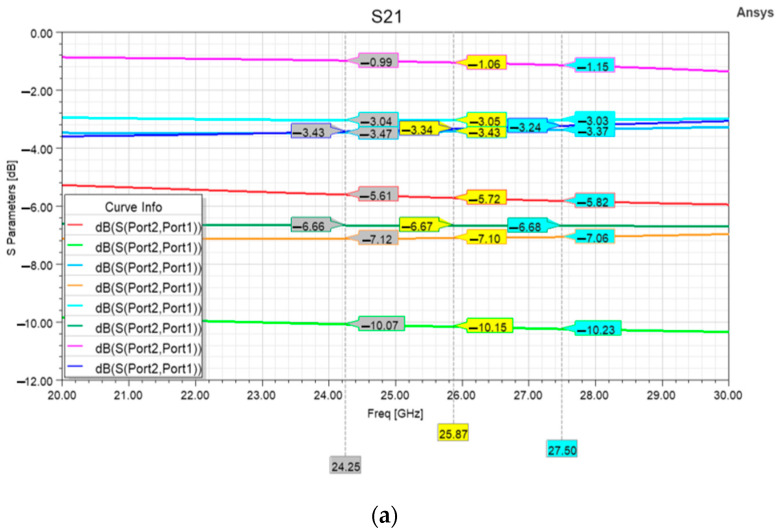
(**a**) Attenuation levels introduced by the device in all its states, and (**b**) return loss curves of the different states. The color of the S21 and S11 curves of a specific state is the same.

**Figure 9 micromachines-15-00895-f009:**
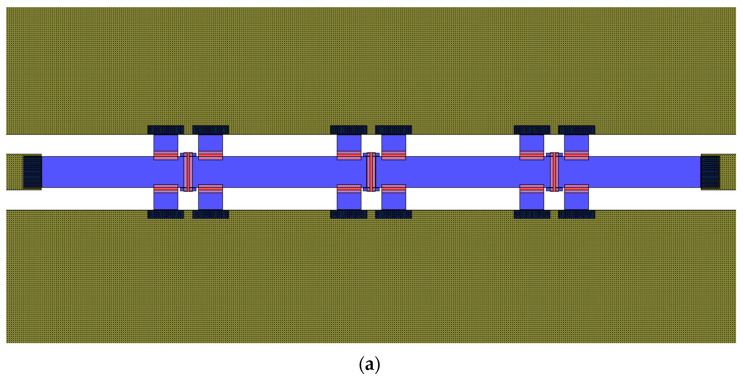
(**a**) Layout of the CPW transmission line loaded with three cascaded cells of optimized Pi-shaped resistors, aiming at an attenuation level of −45 dB. Return loss (red) and attenuation (green) curves in the (**b**) comparison between the measurement outcomes of the plain CPW and the CPW loaded with a single −15 dB Pi resistor, and in the (**c**) comparison between the measurement outcomes of CPW samples loaded with two and three Pi resistors.

**Figure 10 micromachines-15-00895-f010:**
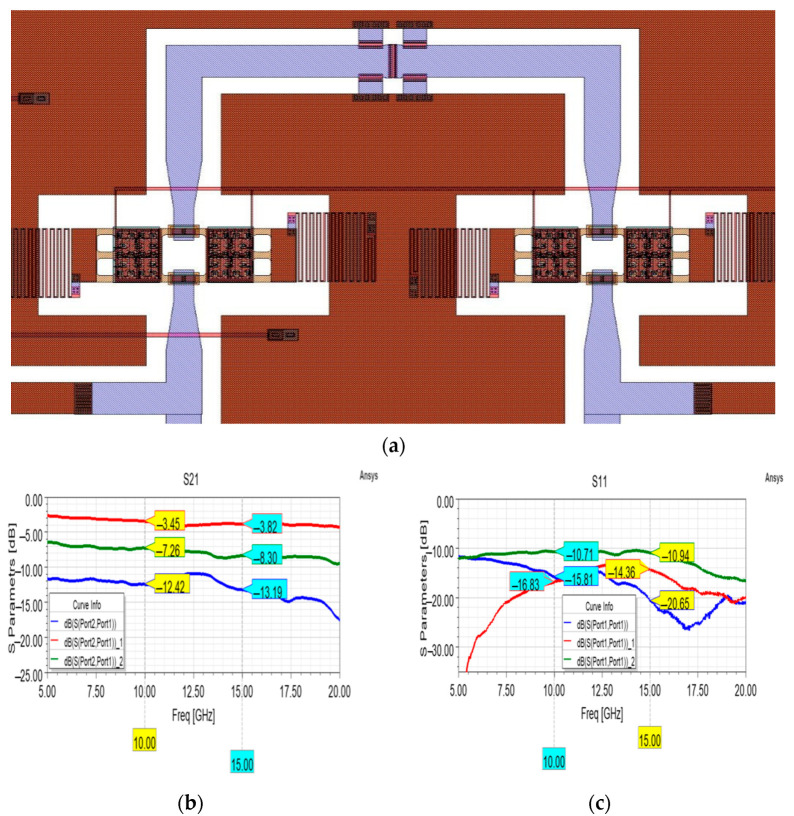
(**a**) Partial layout of the attenuation cell featuring two branches, detail of the loaded branch. Measured (**b**) insertion loss of the unloaded branch (in red) and attenuation levels of both the loaded branch (in blue) and the equivalent parallel attenuation (in green), and the (**c**) related return losses.

**Figure 11 micromachines-15-00895-f011:**
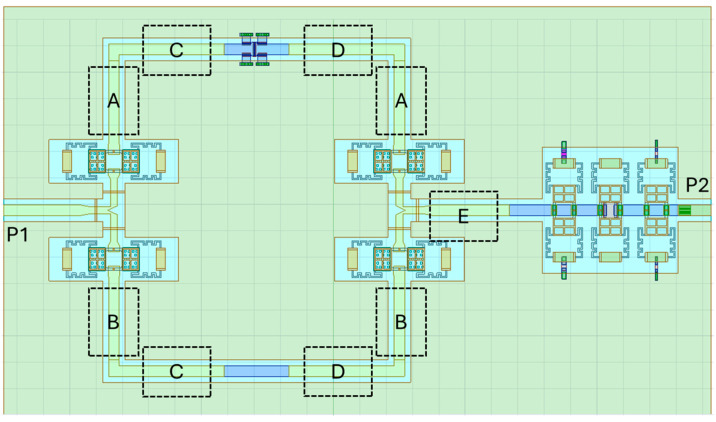
Layout, by principle, of the proposed complete variable attenuator. The dotted boxes identify the sections of CPW lines considered for the final optimization.

**Figure 12 micromachines-15-00895-f012:**
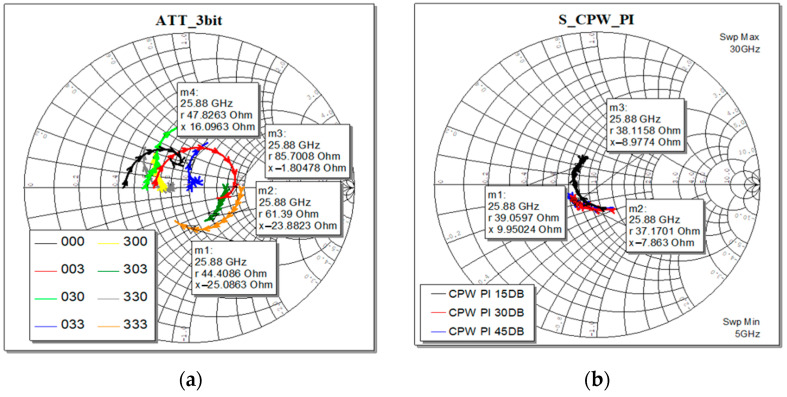
Smith chart representation of the simulated input impedances: (**a**) all eight states of the 3-bit cell variable attenuator; (**b**) the three designed Pi-shaped fixed attenuation CPW section.

**Figure 13 micromachines-15-00895-f013:**
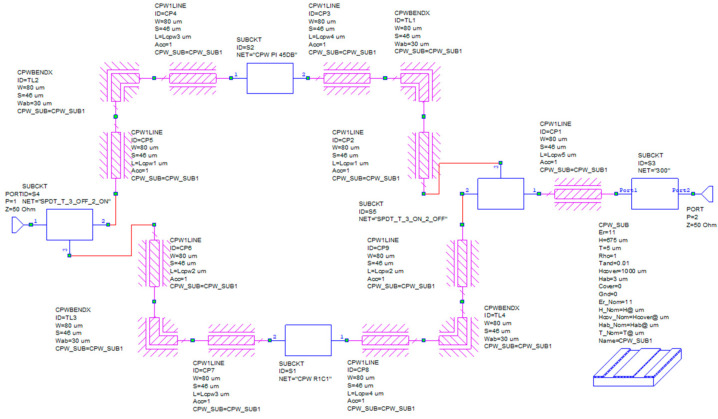
Schematic of the proposed complete variable attenuator in Cadence AWR Design Environment. This figure represents the configuration where the Pi-loaded CPW section is selected by the two SPDTs and the 3-bit cell is set at a state with a maximum attenuation of −10.15 dB.

**Figure 14 micromachines-15-00895-f014:**
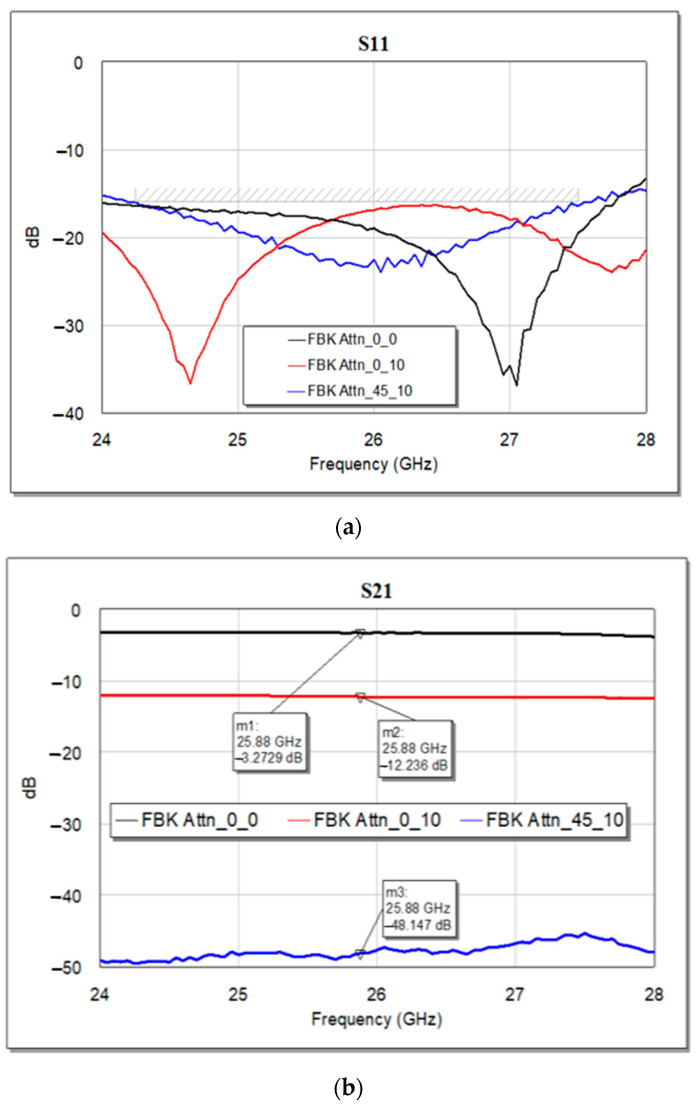
Optimized configuration for the complete attenuator in both its (**a**) return loss and (**b**) insertion loss curves, in all the three selected attenuation states: Case 1 (in black), Case 2 (in red), and Case 3 (in blue).

**Table 1 micromachines-15-00895-t001:** Dimensional features of the discussed class of membranes.

Parameter	Value [µm]	Parameter	Value [µm]
A	35	B	50
C	25	D	15
E	20	F	55
G	380	H	170
J	780	K	270

**Table 2 micromachines-15-00895-t002:** S-Parameters of the 3-bit attenuator when the RF signal line is loaded (1) or not loaded (0) by the resistors of the different attenuation cells, with reference to the arrangement of Figure 4b.

Shunt Left	Series	Shunt Right	Color	S21 at 25.87 GHz [dB]	S11 at 25.87 GHz [dB]
0	0	0	Pink	−1.06	−15.72
0	0	1	Aqua	−3.05	−11.59
0	1	0	Navy blue	−3.34	−11.60
0	1	1	Dark Green	−6.67	−12.68
1	0	0	Blue	−3.43	−9.35
1	0	1	Red	−5.72	−15.49
1	1	0	Orange	−7.10	−10.14
1	1	1	Green	−10.15	−11.67

## Data Availability

Data are contained within the article.

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
