# Peer review of "Discussion and Demonstration of RF-MEMS Attenuators Design Concepts and Modules for Advanced Beamforming in the Beyond-5G and 6G Scenario—Part 2"

_micromachines, 2024, doi:10.3390/mi15070895_

Round 1

Reviewer 1 Report

Comments and Suggestions for Authors

1. The figures in the article should be drawn on certain drawing software, rather than directly captured.

2. The annotation of the result graph should be differentiated, as shown in Figure 8.

3. Please explain why the trend of the pink curve in Figure 8 (b) is different from others.

4. This article should add some corresponding theoretical formulas for the design of structure.

5. The frequency range of the attenuator mentioned in this article is between 24.25-27.5 GHz, but this frequency range is not included in the simulation results in Figure 10. Please explain the reason.

Author Response

Comment 1:  The figures in the article should be drawn on certain drawing software, rather than directly captured.

Response 1: The figures displaying the layouts are clear and come from common software tools used for designing and analyzing RF-MEMS and RF microelectronics devices. The equivalent circuits are drawn with standard symbology. The numerical results are reported in standard graphics, either cartesian or polar form.

Comment 2: The annotation of the result graph should be differentiated, as shown in Figure 8.

Response 2: The result graphs are all accompanied by their own legend, indicating the correspondence between the color of the curve and the related data. When needed, additional comments are in the caption or in the text.

Comment 3: Please explain why the trend of the pink curve in Figure 8 (b) is different from others.

Response 3: 

A specific comment has been added to the manuscript for explaining the behavior of the pink curve and related data:

“Among the different considered states, it is worth mentioning the state in which an attenuation is introduced by the sole insertion loss of the device (000 state in Table 2). In this state, the central resistor is short circuited by the actuated membrane, while the shunt membranes are in rest position. In this situation, the only losses experienced by the signal are due to the small series contact resistance introduced by the actuated central membrane, summed to the capacitive coupling (towards the ground planes) introduced by the two resting shunt membranes. Due to the frequency independent behavior of the contact resistance (that mainly affect the insertion loss of devices), in the 000 state the device can be modeled as a single equivalent shunt switch in rest position, whose return loss depends only on the up-state capacitance. In fact, the slope of the return loss curve of the 000 state in Fig. 8 (b) agrees with the one of a shunt switch in rest position [40].”

Comment 4: This article should add some corresponding theoretical formulas for the design of structure.

Response 4: A comment concerning the design of the 3-bit attenuator has been added, specifying the lack of specific theoretical formulation concerning the design of such switched attenuation modules in the literature. Due to the very nature of the proposed design, its modeling should be done according to the basic model of a loaded lossy transmission line, as detailed in the comment:

“Concerning the theoretical foundation of the device developed by combining such attenuation cells, it worth noticing that no specific formulation devised so far applies to the present case. In fact, if the design rules for fixed Pi-shaped resistors on the basis of the desired attenuation level have been outlined in [29][30], the case of individually switched resistors forming an overall Pi-shaped configuration introduces additional reactive phenomena that are not taken into account in the abovementioned formulation. Therefore, the present layout can be considered as a generic lossy transmission line (widely treated in [39]), loaded by (both series and shunt) switched resistive loads and variable capacitors accounting for the capacitive couplings introduced by the membranes in rest position.”

Conversely, the adopted reference formulas for the design of Pi-shaped resistors have been reported in the following lines:

“a basic cell featuring Pi-shaped resistors inserted along a plain CPW has been designed on the basis of the following design guidelines outlined in [29], dimensioning the series resistor (Rseries) and each of the four shunt resistors (Rshunt) for an attenuation level of -15 dB.

*EQUATION* (1)

*EQUATION* (2)

After an initial simulation, an optimization aimed at achieving a more precise attenuation level of -15 dB was performed, and the devices featuring the optimized resistors were fabricated and characterized.”    

Comment 5: The frequency range of the attenuator mentioned in this article is between 24.25-27.5 GHz, but this frequency range is not included in the simulation results in Figure 10. Please explain the reason.

Response 5: A clarification has been added to the manuscript:

“In fact, as known from the scientific literature [43,44], geometrical discontinuities as the T-junctions and 90° bends should be carefully designed to minimize the impact of the undesired reactive and frequency-dependent phoenomena (increased inductance, capacitive couplings, and thus losses) on the global behavior of the devices. In case of the device reported in Fig. 10 (a), such discontinuities have not been optimized, determining a fluctuating behavior of the attenuation curves of Fig. 10 (b) above 20 GHz. For this reason, the performance of the device has been assessed along the frequency interval in which the effect of the geometrical discontinuities does not dramatically impact the overall behavior of the device.”

Reviewer 2 Report

Comments and Suggestions for Authors

Overall, the paper makes a good impression. Multibit attenuators are in demand in beamforming circuits in new generation communication systems.

It would be nice to cover the following aspects in the article:

- How do the results of circuit modeling compare with the results of electromagnetic simulation? The circuit presented seems too simple (Figure 4), which may not reflect some parasitic effects, parasitic couplings between lines, impedance steps, losses, and etc.

- for the layout presented in Figure 7 there is not enough equivalent circuit to understand how it works.

- in the CPW models in AWR the losses conductor equivalent to gold  are assumed, i.e. Rho=1. In fact this is not true? So please comment that.

Author Response

Comment 1: How do the results of circuit modeling compare with the results of electromagnetic simulation? The circuit presented seems too simple (Figure 4), which may not reflect some parasitic effects, parasitic couplings between lines, impedance steps, losses, and etc.

Response 1: The circuit has been used as a demonstration for the working principle of the device (i.e., the attenuator cell). By purpose, all the parasitics, for sure existent, have been not considered at this stage, since their exact contributions can be evaluated only by full-wave simulations. A specific comment has been added to the manuscript:

“The effects in terms of parasitic impedances due to the geometry of the MEMS structures are not gathered, by purpose, in these simulations, that are devoted to model and analyze the behavior of the series or shunt resistors configured with the CPW lines.”

“Since the beginning of the overall design, the final device is composed of multiple cells, where the effects of parasitic impedances and eventual coupling effects can be modeled only by full-wave simulations of the entire device. The conceptual design and the global optimization of the complete design is described in Section 5.

Comment 2: for the layout presented in Figure 7 there is not enough equivalent circuit to understand how it works. 

Response 2: The equivalent circuit has been drawn in Figure 7c. The circuit describes the working principle of the device, following the same approach described for the basic attenuator cell.

Comment 3: in the CPW models in AWR the losses conductor equivalent to gold are assumed, i.e. Rho=1. In fact this is not true? So please comment that.

Response 3: The realized devices have a 5 µm thick electroplated gold. So, the skin effect can be neglected at the operating frequencies of the device. For this reason, we can consider in the simulation the conductivity of the metal conductors equivalent to the gold one.